# Opposing Roles of apolipoprotein E in aging and neurodegeneration

Eloise Hudry[1], Jacob Klickstein[1], Claudia Cannavo[2], Rosemary Jackson[2], Alona Muzikansky[3], Sheetal Gandhi[1], David Urick[1], Taylie Sargent[1], Lauren Wrobleski[1], Allyson D Roe[1], Steven S Hou[1], Kishore V Kuchibhotla[4], Rebecca A Betensky[3], Tara Spires-Jones[2], Bradley T Hyman[1]

**Apolipoprotein E (APOE) effects on brain function remain controversial. Removal of APOE not only impairs cognitive functions but also reduces neuritic amyloid plaques in mouse models of Alzheimer's disease (AD). Can APOE simultaneously protect and impair neural circuits? Here, we dissociated the role of APOE in AD versus aging to determine its effects on neuronal function and synaptic integrity. Using two-photon calcium imaging in awake mice to record visually evoked responses, we found that genetic removal of APOE improved neuronal responses in adult APP/PSEN1 mice (8–10 mo). These animals also exhibited fewer neuritic plaques with less surrounding synapse loss, fewer neuritic dystrophies, and reactive glia. Surprisingly, the lack of APOE in aged mice (18–20 mo), even in the absence of amyloid, disrupted visually evoked responses. These results suggest a dissociation in APOE's role in AD versus aging: APOE may be neurotoxic during early stages of amyloid deposition, although being neuroprotective in latter stages of aging.**

## Introduction

Apolipoprotein E (APOE) in the central nervous system (CNS) has been the focus of study for two reasons: it is the most abundantly expressed apolipoprotein in the CNS (Bjorkhem & Meaney, 2004; Huang & Mahley, 2014), and inheritance of the E4 allele of the *APOE* gene profoundly impacts the risk for Alzheimer's disease (AD), exacerbating amyloid deposition and worsening cognition and synapse loss. We, therefore, sought to examine the effect of APOE depletion on neuronal function and synaptic integrity in adult or aged mice in both physiological and pathophysiological contexts.

As the primary CNS apolipoprotein, APOE is responsible for much of the regulation of the brain lipid metabolism, particularly the transfer of cholesterol and phospholipids from glial cells to neurons (Boyles et al, 1985; Pitas et al, 1987; Pfrieger & Ungerer, 2011). During adulthood, neurons rely on cholesterol from glial cells for many processes; thus APOE plays an important role in modulating synapse growth, stabilization, and renewal in a physiological context (Holtzman & Fagan, 1998; Mauch et al, 2001). APOE is also involved in removing cholesterol and lipids from the CNS, therefore controlling the clearance of cellular debris and promoting remyelination in the aged CNS and some neurodegenerative diseases (Mahley, 1988; Zlokovic, 2011; Cantuti-Castelvetri et al, 2018). Other functions of APOE in the neural tissue include buffering oxidative stress (Evola et al, 2010; Chen et al, 2015) and preserving the integrity of the blood–brain barrier (Fullerton et al, 2001; Hafezi-Moghadam et al, 2007; Nishitsuji et al, 2011), further emphasizing the pivotal role of APOE in maintaining brain homeostasis.

Previous studies have observed that a complete lack of APOE in murine models leads to cognitive impairment when compared with wild-type mice (Gordon et al, 1995; Masliah et al, 1997; Kitamura et al, 2004; Trommer et al, 2004; Yang, Gilley et al, 2011a; Zerbi et al, 2014), whereas others failed to detect similar deficits (Hartman et al, 2001; Bour et al, 2008). A recent case study of a 40-yr-old man with a complete absence of *APOE* expression initially reported normal cognitive function (despite dramatic hypercholesterolemia [Mak et al, 2014]), but a second in-depth evaluation showed some evidence of cognitive impairment (Cullum & Weiner, 2015). Whether these discrepancies result from the use of different cognitive tasks or from the age of the animals and subjects included in each study is unclear, but there is no doubt that further investigation of the specific impact of APOE on neuronal function in vivo remains an important unmet goal.

In the context of disease, *APOE* was identified more than two decades ago as a significant modulator of the risk for late-onset AD (Wisniewski & Frangione, 1992; Corder et al, 1993, 1994; West et al, 1994; Hyman et al, 1996; Lippa et al, 1997). APOE is a well-established partner of amyloid β (Aβ) peptides, catalyzing Aβ oligomerization, aggregation in the parenchyma (Holtzman et al, 2000; Fagan et al, 2002; Hashimoto et al, 2012), clearance (Deane et al, 2008;

[1]Alzheimer Research Unit, Department of Neurology, Massachusetts General Hospital and Harvard Medical School, Charlestown, MA, USA   [2]Centre for Discovery Brain Sciences, UK Dementia Research Institute, and Edinburgh Neuroscience, The University of Edinburgh, Edinburgh, UK   [3]Department of Biostatistics, Harvard T.H. Chan School of Public Health, Boston, MA, USA   [4]Johns Hopkins University, Baltimore, MD, USA

Correspondence: ehudry@mgh.harvard.edu

Castellano et al, 2011; Hudry et al, 2013), and recruitment to the synapse (Koffie et al, 2012). More recently, APOE has also been identified as a molecular trigger of the amyloid-dependent neuroinflammatory response via its role as a ligand for the triggering receptor expressed on myeloid cells 2 (TREM2) (Atagi et al, 2015; Yeh et al, 2016). Disruption of the murine *apoE* gene in AD transgenic models significantly delays the formation of the so-called "dense core" Thio-S–positive amyloid plaques (Bales et al, 1997; Irizarry, Cheung et al, 2000a), even though substantial load of diffuse amyloid and elevated concentrations of soluble A$\beta$ peptides remain in the parenchyma (Irizarry, Rebeck et al, 2000b). These results suggest that a complete lack of APOE may have a beneficial impact on amyloidopathy, a hypothesis recently validated using an approach by antisense oligonucleotide-based knockdown of *APOE* in mouse models of amyloidosis (Huynh et al, 2017). However, the consequences of *APOE* genetic disruption on neuronal function and synaptic integrity are still being debated. The question remains if APOE can simultaneously protect and impair brain homeostasis. The present study aims to examine both APOE's role in normal physiology and in A$\beta$-induced neurotoxicity. To do so, we investigate how APOE affects neuronal function and synaptic integrity rather than only focusing on amyloid changes.

In this study, we used two-photon calcium imaging in the visual cortex to measure visually evoked neuronal responses (Andermann et al, 2011; Grienberger et al, 2012; Kuchibhotla et al, 2014) and array tomography to assess synapse density at a single-synapse resolution (Koffie et al, 2012; Tai et al, 2012; Kay et al, 2013). We systematically evaluated the impact of the presence or absence of APOE on neuronal function and synaptic integrity in mice that develop plaques (APP/PSEN1 mice expressing both the human mutated Amyloid precursor protein, APP, and presenilin-1, PSEN1,

genes) and those without increased amyloidosis (wild-type). Our results demonstrate that in the context of amyloid pathology, APOE enables A$\beta$-dependent neuronal dysfunction and synaptotoxicity and a dramatic protective effect is observed by ApoE[null]. By contrast, APOE also appears as an important factor to preserve brain function during aging, even in the absence of amyloid deposition. These findings, therefore, dissociate APOE's role towards amyloid neuropathological changes versus normal aging and warrant further consideration of the impact of APOE on neuronal function in addition to its effect on amyloid.

# Results

### Disruption of visually evoked responses in transgenic mice model of amyloidosis

To establish how APOE modulates neuronal function and amyloid-dependent dysfunction in vivo, we recorded neuronal calcium transients triggered by visual stimulation in wild-type, APOE[null], APP/PSEN1, and APP/PSEN1/APOE[null] mice (Fig 1A). The study of neuronal dysfunction in the visual cortex is relevant to AD, as deficits in central sensory processing have been reported in the disease, particularly at advanced stages (Cronin-Golomb et al, 1991; Bublak et al, 2011). The visual area V1 is also easily accessible for intra-vital calcium imaging (Andermann et al, 2011) and constitutes the output of a relatively simple circuitry downstream of the retina and the lateral geniculate nucleus within the thalamus (Seabrook et al, 2017), therefore facilitating the recording of neuronal responses to well-controlled sensory stimuli. In addition, two age groups were included in our study (8–10-mo-old "adult" mice and

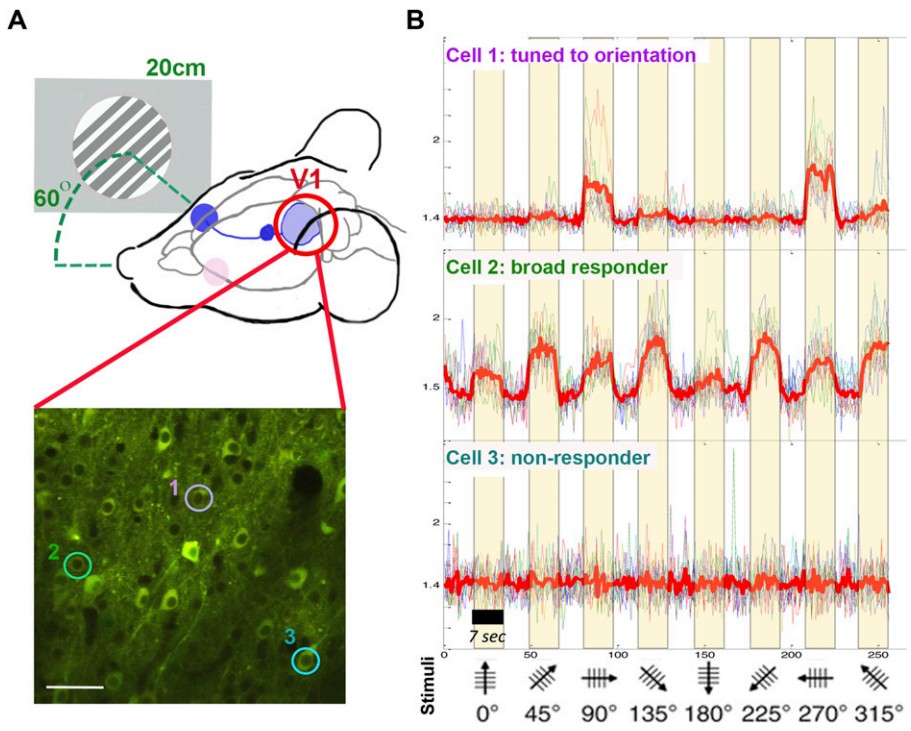

**Figure 1. Experimental setting for in vivo recording of visually evoked responses.**
**(A)** Calcium transients were recorded in the primary visual cortex V1 while presenting eight different drifting gratings to the contralateral eye (screen–eye distance, 20 cm; screen-midline angle, 60 Åā). The lower panel shows a representative in vivo two-photon image of AAV-CBA-YC signal detected in neurons one month after AAV stereotaxic injection. **(B)** Examples of averaged traces for three neurons individually selected in (A) show that the responses are highly heterogeneous within the same field of view, from nonresponsive (cell #3) to broadly responsive (cell #2) and neuron responding selectively to a particular orientation of the stimulus (cell #1). Bar graph: 100 $\mu$m.

18–20-mo-old "aged" mice) to determine the impact of age, genotype, and the interaction of both parameters (Fig S1A). We could record robust neuronal responses to the visual stimuli in all experimental groups. In our stimulation protocol, responding cells were detected in the primary visual area (mean = 6.89 ± 1.05% for wild-type littermates; 13.76 ± 2.13% for the APP/PS1; 9.75 ± 2.3% for the APOE[null]; and 10.58 ± 1.53% for the APP/PS1/APOE[null] mice, Fig S1A and B). Within the same field of view, a wide range of different response patterns was observed, including nonresponsive cells (no correlation between calcium transients and visual stimulation), broadly responsive cells (increased firing simultaneously to any stimulus), and visually tuned neurons responding to a specific orientation and/or direction of the visual stimulus (Fig 1B). There was no statistical difference between the percentage of "responding cells" among the experimental groups (Fig 2A, P = 0.0682), with variability depending whether or not the region of interest (ROI) considered was exactly located within the primary visual cortex area or at the edge of it (Fig S2).

When establishing our protocol, we observed that in contrast to the broadly tuned cells that responded to any stimuli, occasional neurons systematically fired when the stimuli went off (which we defined as "off-responders," Fig 2B), a phenomenon previously described when recording visual responses in awake animals (Jin et al, 2008; Liang et al, 2008). When we calculated the percentage of "off-responding" neurons in comparison with the overall number of responding cells within the same fields of view, we observed a significant increased proportion of "off-responders" specifically in APP/PSEN1 mice (16.57 ± 4.68%, Kruskal–Wallis test, P = 0.0068, Fig 2C), which tended to be exacerbated in aged as compared with adult transgenic APP/PSEN1 animals (even though the difference

between "adult" and "aged" APP/PSEN1 did not reach statistical significance, Fig S3). Intriguingly, the percentage of off-responding neurons was much lower in all the other groups considered (wild-type: 4.32 ± 1.27%; APOE[null]: 4.68 ± 1.19%; and APP/PSEN1/APOE[null]: 3.2 ± 0.84%), and especially in APP/PSEN1/APOE[null] mice lacking *apoe* expression, suggesting that APOE[null] mice normalizes an alteration from normal physiology observed in APP/PSEN1 mice.

*Apoe* genetic ablation in adult APP/PSEN1 mice restores visual selectivity but sensitizes the brain towards age-associated neuronal dysfunction.

To gain further insight into the functional integrity of the visual network between wild-type, APOE[null], APP/PSEN1, and APP/PSEN1/APOE[null] mice, we then compared the direction selectivity index (DSI) and orientation selectivity index (OSI) between these groups (Fig 3A and B). DSI and OSI represent quantifiable measurements of the ability of visual neurons to respond to a principal orientation or direction of a visual stimulus as compared with others, and higher DSI and OSI values correspond to a more specific tuning to the stimulation. Of importance, this functional feature (the tuning of visual neurons to stimulation) has been previously reported to be impaired in mouse models of amyloidopathy (Grienberger et al, 2012). After fitting a linear mixed model with "genotype" and "age" as fixed effect and "mouse" as random effect, we demonstrated that the log-adjusted DSI and OSI were significantly associated with the genotype (P = 0.0013 for DSI and P = 0.0076 for OSI) and with the age of the mice (P = 0.0325 for DSI and P = 0.0068 for OSI). In particular, the average of both DSI and OSI was lower in APP/PSEN1 mice as compared with wild-type animals (P = 0.0002 and P = 0.0008, respectively), therefore demonstrating impaired tuning to visual

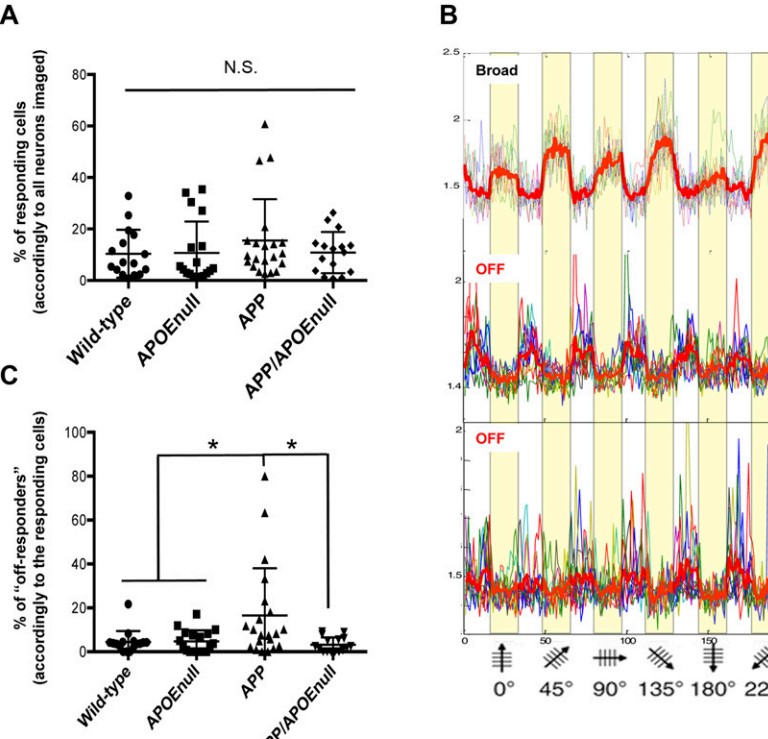

**Figure 2. "Responders" and "off-responders" proportions.**
**(A)** Scatter dot plot representing the percentage of neurons responding to visual stimulation among all neurons analyzed in each animal. **(B)** Representative traces or visually evoked neuronal responses showing the concomitant presence of broad and off-responding cells in the same field of view during the same imaging session. **(C)** Scatter dot plot of the percentage of "off-responding" cells across genotypes (calculated accordingly to the number of responsive neurons within the same fields of view). n = 18 wild-type, n = 13 APOE[null], n = 18 APP/PSEN1, and n = 16 APP/APOE[null] mice (adult and old cohorts); Kruskal–Wallis test followed by post hoc Dunn's Multiple comparison test; *P < 0.05.

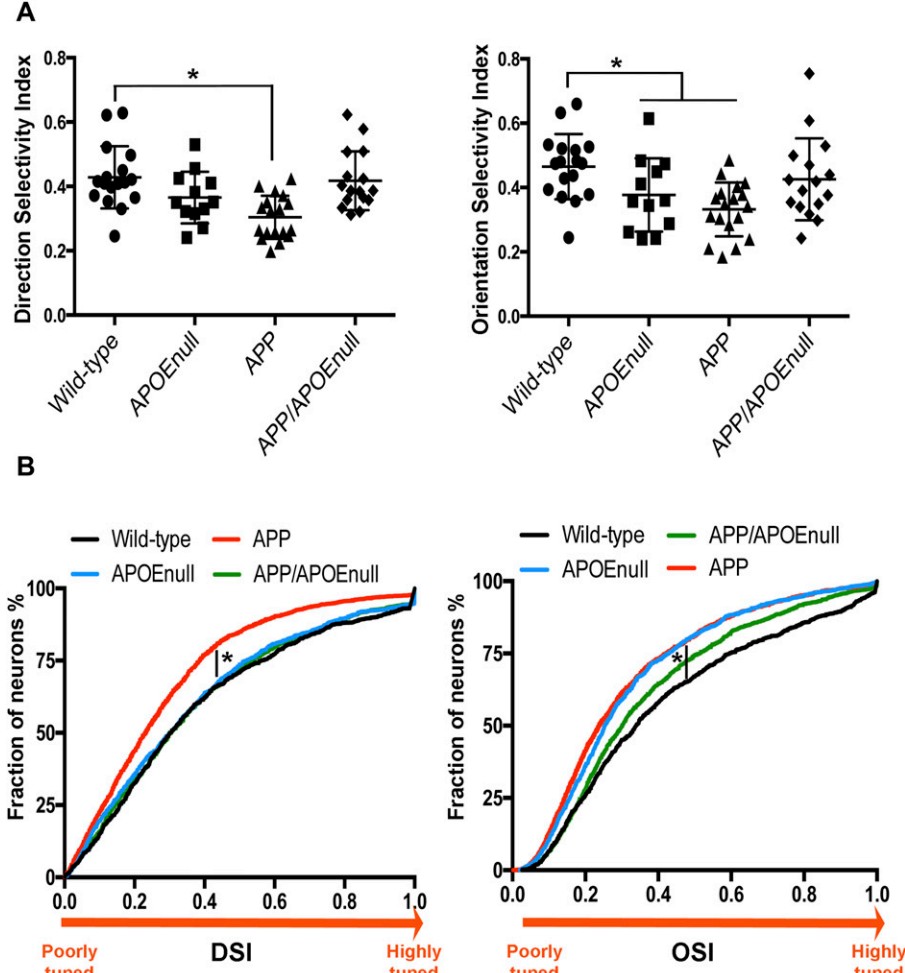

**Figure 3. Tuning to visual stimuli in wild-type, APOE[null], APP/PSEN1, and APP/PSEN1/APOE[null] mice. (A)** Scatter dot plots representing the DSI (left panel) and OSI (right panel) calculated after recording visually evoked response in wild-type, APOE[null], APP/PSEN1, and APP/PSEN1/APOE[null] mice. **(B)** Cumulative distributions of DSI and OSI calculated for all responding neurons showing a shift towards an increased proportion of poorly tuned neurons in APP/PSEN1 as compared with the other genotypes, with the exception of the OSI distribution in APOE[null] mice. n = 18 wild-type, n = 13 APOE[null], n = 18 APP/PSEN1, and n = 16 APP/APOE[null] mice (adult and old cohorts); linear mixed model after log-transformation of the data; *$P < 0.05$.

stimulation in this model of amyloidosis. This was not the case between the control and APP/PSEN1/APOE[null] groups ($P = 0.4091$ and $P = 0.1584$). These results, therefore, demonstrated that the absence of APOE largely abrogates A$\beta$-dependent neuronal dysfunction in a mouse model of amyloidosis. However, APOE[null] mice still showed a mild decreased OSI average when compared with wild-type controls ($P = 0.0294$), implying that the complete lack of APOE itself can impact tuning performances.

Pair-wise comparisons between adult and aged mice (Fig S4A) revealed more prominent age-dependent decreases of the OSI values in APP/PSEN1/APOE[null] ($P = 0.037$), with a similar but non-significant decrease in APOE[null] ($P = 0.109$) mice. No age-dependent change was detected for the control or APP/PSEN1 groups, which respectively presented with high or low OSI means across the lifespan of the mice. The DSI values showed fewer differences with only a significant decreased average of the DSI detected in aged APP/PSEN1 mice ($P = 0.0445$). The higher sensitivity of OSI to discriminate changes in visual tuning can be explained by the fact that the OSI was calculated based on the tuning discrimination between the "favorite" orientation as compared with three other orientations, whereas the DSI was calculated based on preferential firing for one direction as compared with the opposite one. When compared with age-matched

littermates (Fig S4B), both adult and aged APP/PSEN1 mice were significantly impaired for OSI and DSI ($P = 0.011$ for DSI and $P = 0.0012$ for OSI when comparing adult APP/PSEN1 and littermates; and $P = 0.022$ for DSI and $P = 0.042$ for OSI when comparing aged APP/PSEN1 and littermates). However, only the OSI values were significantly diminished in aged APOE[null] mice as compared with aged controls ($P = 0.04$). The averaged OSI in aged APP/PSEN1/ApoE[null] mice was also lower than wild-type animals, but this difference did not reach significance ($P = 0.084$). These results demonstrated early and pronounced neuronal dysfunction in mice model of A$\beta$ amyloidosis, a phenotype significantly improved in adult mice lacking APOE. However, age-dependent deficits were also detected in older APOE[null] and, to a lesser extent, APP/PSEN1/APOE[null] mice. We concluded that APOE is a necessary factor participating in amyloid-dependent loss of network integrity, although complete lack of APOE independently triggers mild neuronal dysfunction with age.

### *Apoe* genetic ablation does not impact global amyloid burden but affects the aggregation state of A$\beta$

To evaluate how the "rescue effect" observed in mice lacking APOE correlated with the level of amyloid pathology, we performed a

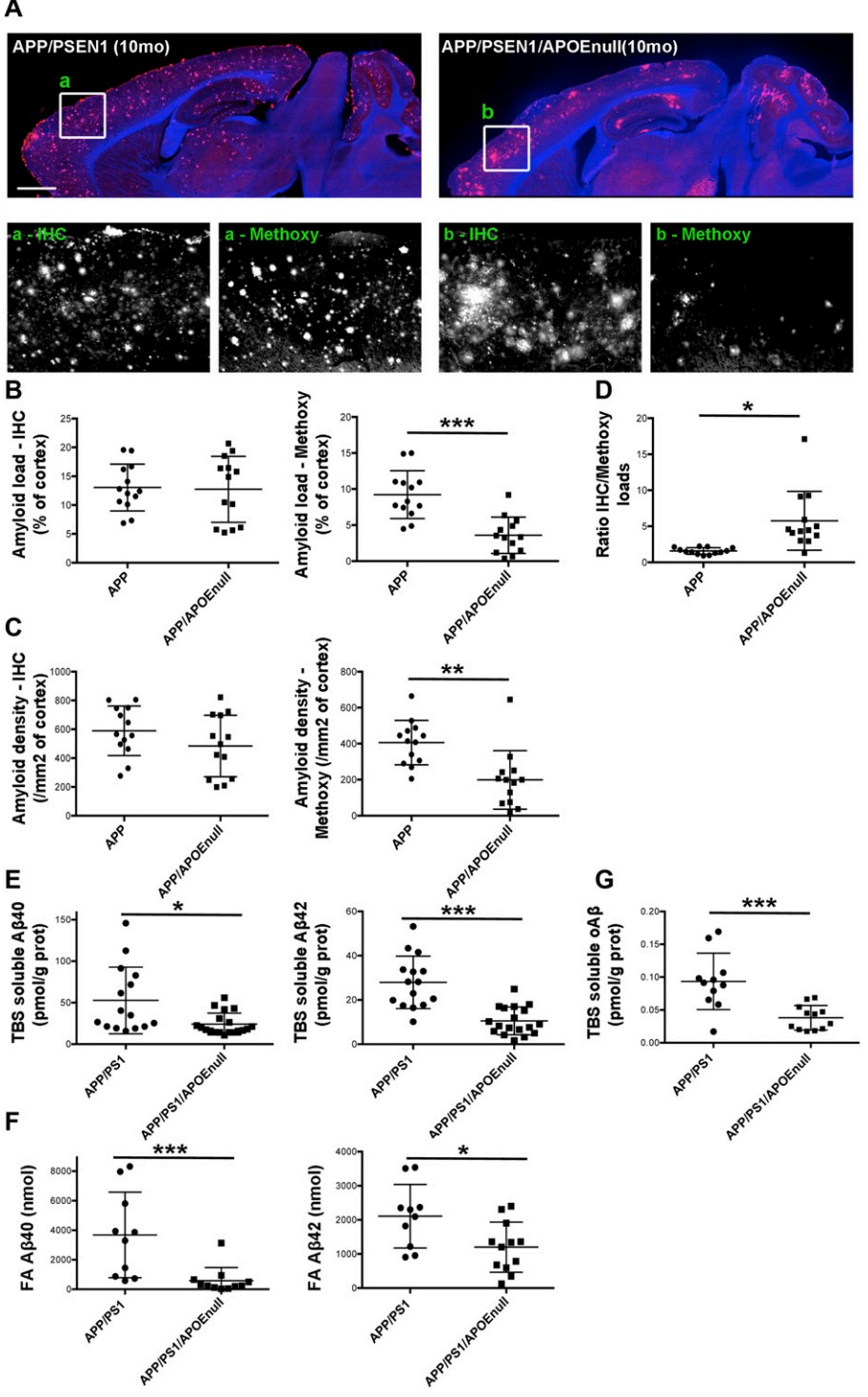

**Figure 4. Amyloid pathology in APP/PSEN1 and APP/PSEN1/APOE^null mice.**

**(A)** Representative images of the global amount of amyloid in the cortex of APP/PSEN1 and APP/PSEN1/APOE^null mice at 10-mo old after anti-Aβ immunostaining (IH). As shown in the lower panels, the overall load of amyloid was comparable between genotypes, but the amount of dense-core plaques detected with Methoxy-XO4 was greatly diminished in the absence of APOE. **(B)** Scatter dot plot representing the stereological quantification of amyloid load in APP/PSEN1 and APP/PSEN1/APOE^null animals (including both adult and aged cohorts) after anti-Aβ immunostaining or Methoxy-XO4 labeling. **(C)** Scatter dot plots representing the stereological quantification of plaque density in APP/PSEN1 and APP/PSEN1/APOE^null animals. **(D)** The ratios of IH/Methoxy calculated per mouse showed that the amyloid deposits are mostly diffuse in APP/PSEN1/APOE^null mice as compared with APP/PSEN1. **(E)** The biochemical quantification of the concentrations of TBS-soluble Aβ40 and Aβ42 revealed lower levels of Aβ peptides in mice devoid of APOE as compared with APP/PSEN1 mice (both age groups pooled together). **(F)** A parallel result was observed when quantifying the levels of FA Aβ40 and Aβ42. **(G)** A similar difference was observed when the concentration of Aβ oligomers was measured from TBS brain extracts. Scale bar = 1,000 μm. n = 10 to 18 mice/group; unpaired t test; *P < 0.05, **P < 0.001, ***P < 0.0005.

stereological analysis of the load and density of amyloid plaques in the cortex of APP/PSEN1 and APP/PSEN1/APOE^null mice. Conventional anti-Aβ immunolabeling detecting all amyloid aggregates revealed that neither the burden nor the density of deposits was significantly different between APP/PSEN1 and APP/PSEN1/APOE^null (Fig 4A–C). By contrast, a dramatic decrease in the amount of

fibrillar deposits stained by Methoxy-XO4 was observed in mice lacking APOE (Fig 5D; P < 0.0001 and P = 0.0004, respectively, for Methoxy load and density between APP/PSEN1 and APP/PSEn1/APOE^null mice), in agreement with previous reports showing that APOE significantly affects the aggregation state of Aβ neurotoxic species (Irizarry et al, 2000a; Holtzman et al, 2000; Fagan et al, 2002).

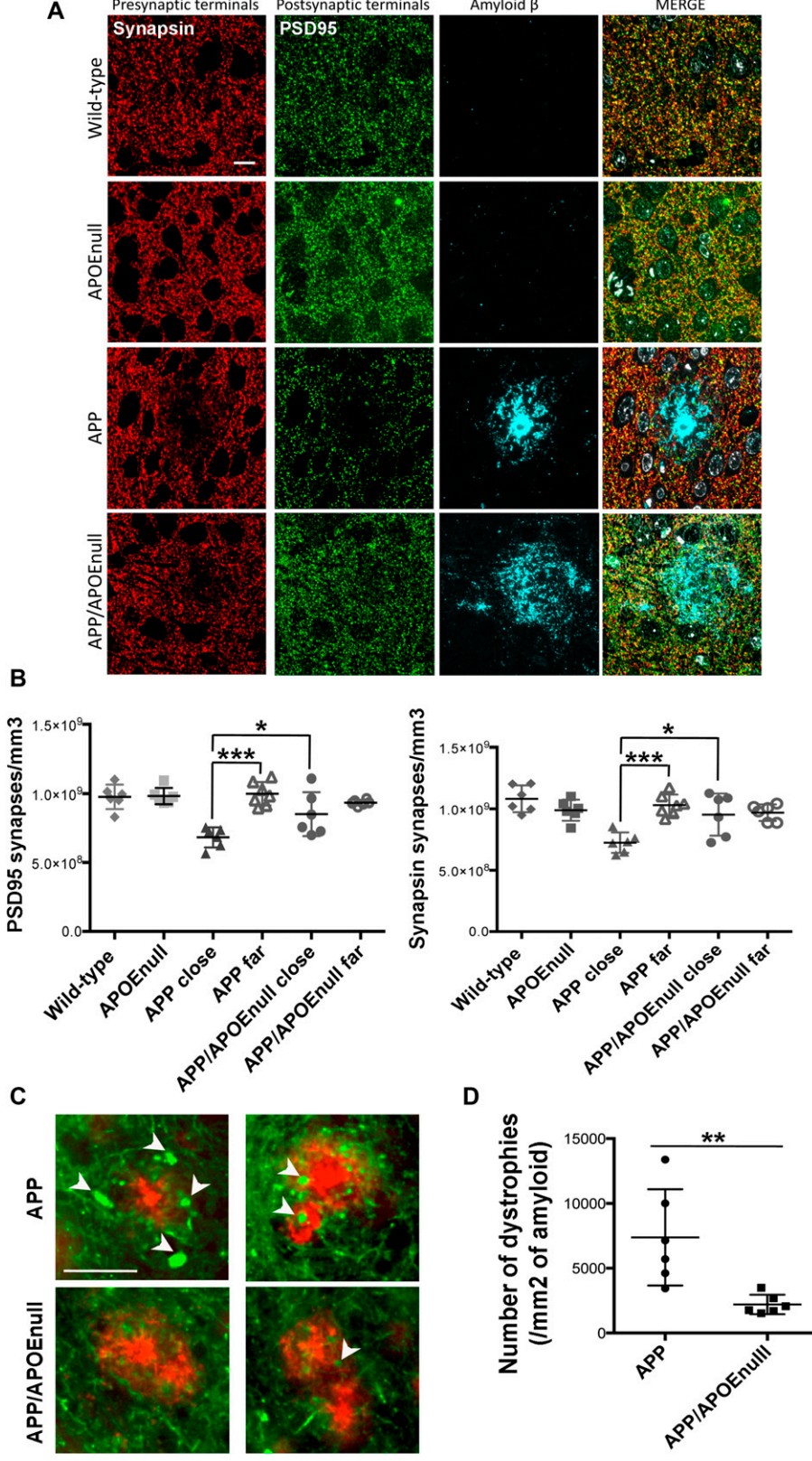

**Figure 5. Ab-dependent synaptic loss and neuritic dystrophies in adult APP/PSEN1 and APP/PSEN1/APOE[null] mice.**

**(A)** Representative images of array tomography ribbons after immunostaining for Ab (1C22), presynaptic terminals (antisynapsin I, red), postsynaptic densities (anti-PS95, green), and nuclei (DAPI, blue). It is possible to appreciate how diffuse is the amyloid in the brain of APP/APOE[null] mouse as compared with the compact staining observed in the APP mouse. Scale bar = 10 $\mu$m. **(B)** Scatter dot plots summarizing the density of PSD95 (left panel) and synapsin (right panel)-positive puncta in the cortex of wild-type, APOE[null], APP/PSEN1, and APP/PSEN1/APOE[null] mice. A significant decrease in synaptic density was only observed in the close vicinity of amyloid plaques in AD transgenic mice. n = 6 mice/group (10–12 mo "adult" cohort); Two-way ANOVA and Tukey's post hoc test. *$P < 0.05$ and ***$P < 0.0001$. **(C)** Representative images of neuritic dystrophies (arrows) around amyloid plaques observed in APP/PSEN1 and APP/PSEN1/APOE[null] mice, showing that most sprouting axons are found in mice expressing endogenous murine apoE. Scale bar = 50 $\mu$m. **(D)** Scatter dot plot summarizing the stereological quantification of the number of dystrophies observed per area of amyloid (evaluated by immunohistological staining). n = 6 mice per group (10–12 mo "adult" cohort); Mann–Whitney test. *$P < 0.005$.

Consequently, the ratio of amyloid load between "all deposits" and dense core plaques was significantly higher in APP/PSEN1/APOE[null] as compared with APP/PSEN1 mice (Fig 4D). Within each genotype, the amyloid load progressed between adult and aged mice, but the increase in Methoxy-XO₄–positive plaques was not as prominent as the changes observed when both diffuse and compact deposits were analyzed altogether (Fig S5). To complete this analysis, we determined the concentrations of Aβ40 and Aβ42 in the TBS and formic acid (FA) fractions of the brains, and we observed a general decrease in the amounts of Aβ species in APP/PSEN1/APOE[null] mice as compared with APP/PSEN1 (Fig 4E–G; $P = 0.0169$ and $P < 0.0001$, respectively, for TBS soluble Aβ40 and Aβ42; $P = 0.0004$ and $P = 0.019$, respectively, for FA soluble Aβ40 and Aβ42). Interestingly, these differences were essentially due to the substantial levels of amyloid peptides in aged APP/PSEN1 animals (Fig S6A–C), whereas the amounts of both soluble and insoluble Aβ40/42 were comparable between the adult cohorts (for which the lack of APOE alleviates Aβ-dependent neuronal dysfunction). These data emphasize the weak correlation that exists between the global amount of amyloid and neuronal dysfunction and illustrate the necessity to assess amyloid-dependent neurotoxicity from a functional perspective. No substantial difference in the amount of amyloid pathology was detected between adult APP/PSEN1 and APP/PSEN1/APOE[null] mice (except for the presence of dense-core neuritic plaques), even though the absence of APOE in the later rescued the neuronal responses to visual stimuli. These findings strengthen the rationale for down-regulating *ApoE* expression as a possible therapeutic approach to alleviate Aβ-dependent neuronal impairment.

### *Apoe* genetic ablation does not alter the global levels of excitatory and inhibitory neurotransmitter receptors but rescues synaptic integrity near Aβ plaques

To determine if the functional alterations detected by recording visually evoked neuronal responses in vivo might correlate with changes of the levels of excitatory or inhibitory neurotransmitter receptors at the synapse, we performed a biochemical analysis after preparing synaptoneurosome-enriched fractions from the cortex of adult wild-type, APOE[null], APP/PSEN1, and APP/PSEN1/APOE[null] mice (10–12 mo) (Fig S7). Overall, no significant difference was detected in the levels of *N-methyl-D-aspartate* receptor 1 (NMDAR1), 2A (NMDAR2A), 2B (NMDAR2B), *γ-aminobutyric acid* receptors A (GABAR-A), and B (GABAR-B) between all the experimental groups (Fig S8). These results demonstrate that the neuronal circuitry dysfunction previously observed in APP/PSEN1 mice do not result from a global difference in the levels of excitatory or inhibitory synaptic markers.

Because local loss of synapses is one of the well-established downstream effects of amyloid deposition that may participate in network deterioration (Pozueta et al, 2013; Spires et al, 2005; Spires-Jones & Hyman, 2014; Hong et al, 2016; Arbel-Ornath et al, 2017) and considering that we detected a change in the aggregation state of amyloid deposits in the cortex of APP/PSEN1/APOE[null] mice, we tested the hypothesis that the improved visual tuning observed in the absence of APOE could be directly related with a decreased local synaptotoxicity of Aβ peptides present in plaques in those mice. Using the high-resolution imaging technique array

tomography, the density of pre- (synapsin) and post-synaptic (PSD95) elements was determined in the cortical mantle of the adult cohort previously imaged. For all plaque-bearing animals (APP/PSEN1 and APP/PSEN1/APOE[null]), the synaptic density in areas of the cortex close (<10 µm) and far (>35 µm) from amyloid plaques was independently measured. Total synapsin 1 and PSD95 puncta were used as measurements of synaptic density in the four mouse lines, as previously done in other similar studies (Koffie et al, 2009; Koffie et al, 2012; Kay et al, 2013) (Fig 5A and B). A two-way ANOVA test with distance from the plaque and genotype as variables was used and detected a significant difference across both variables (distance from plaque: $F(1,41) = 15.94$, $P = 0.0003$; genotype: $F(3,41) = 7.658$, $P = 0.0004$). Comparison of individual groups using a Tukey's post hoc test revealed that a 30% synaptic loss was observed near amyloid plaques in APP/PSEN1 mice (as previously observed [Koffie et al, 2009] and Fig 5B), with an averaged density of PSD95 and synapsin puncta close to plaques of $6.83 \times 10^8 \pm 2.76 \times 10^7/mm^3$ and $7.24 \times 10^8 \pm 3.14 \times 10^7/mm^3$, respectively, as compared with $9.99 \times 10^8 \pm 3.21 \times 10^7$ and $10.30 \times 10^8 \pm 3.26 \times 10^7$ synapses/$mm^3$ far from plaques ($P < 0.0001$). This decrease in synaptic density in the direct vicinity of amyloid in APP/PSEN1 mice was also significant compared with wild-type ($9.77 \times 10^8 \pm 3.63 \times 10^7$ PSD95 puncta/$mm^3$ and $10.82 \times 10^8 \pm 4.46 \times 10^7$ synapsin puncta/$mm^3$; $P < 0.0001$) and APOE[null] ($9.83 \times 10^8 \pm 2.42 \times 10^7$ PSD95 puncta/$mm^3$ and $9.89 \times 10^8 \pm 3.54 \times 10^7$ synapsin puncta/$mm^3$, $P < 0.0001$) controls. Interestingly, no drop of the synaptic density was detected close to plaques in APP/PSEN1/APOE[null] mice ($8.51 \times 10^8 \pm 76.51 \times 10^7$ PSD95 puncta/$mm^3$ and $9.53 \times 10^8 \pm 7.03 \times 10^7$ synapsin positive puncta/$mm^3$), which was significantly higher than the synaptic density measured in the vicinity of amyloid deposits in APP/PSEN1 mice ($P = 0.0345$), and comparable with wild-type and APOE[null] animals. These results, thus, imply that the presence of both Aβ and APOE is necessary to observe synaptic collapse in AD transgenic mice and that abolishing *apoE* expression suffices to rescue amyloid-dependent synaptic loss and the functional impairments observed in APP/PS1 mice. Intriguingly, when the accumulation of Aβ at the synapse in APP/PSEN1 and APP/PSEN1/APOE[null] was measured as the percentage of pre- and post-synaptic densities co-localizing with amyloid, no significant difference could be observed between those groups (Fig S9A), thus challenging the concept of APOE acting as a chaperone mediating Aβ accumulation at the synapse. In addition, the synaptic volume decreased significantly close to plaques in APP/PSEN1 mice, a parameter that remained unchanged in the absence of APOE (Fig S9B).

### The absence of APOE decreases Aβ-associated neuritic anomalies and glial reactivity

To complete our evaluation of the impact of APOE on Aβ-dependent neurotoxic effects in the microenvironment of amyloid plaques, we performed a stereological analysis of the density of neuritic dystrophies around each deposit (Fig 5C) and Aβ-dependent glia reactivity in the cortex of APP/PSEN1 and APP/PSEN1/APOE[null] animals (adult cohort). Although numerous abnormal neurites were generally associated with amyloid plaques in APP/PSEN1 mice, they were seldom in APP/PSEN1/APOE[null] animals (Fig 5D, $P = 0.043$). In addition, when we evaluated the number of reactive

astrocytes (Glial fibrillary acidic protein, GFAP, positive) and microglia (Iba1 positive) clustered around amyloid aggregates in APP/PSEN1 and APP/PSEN1/APOE[null] mice, we observed a significant decreased density of both glial cell types in animals lacking APOE (Fig 6, P = 0.0006 for microglia and P = 0.0012 for astrocytes). These results not only establish APOE as an essential modulator of amyloid deposition but also demonstrate its strong impact on a large panel of Aβ-associated neurotoxic events and downstream inflammatory reactions in the local vicinity of amyloid plaques. Whether or not these effects are a direct consequence of the change in the aggregation state and toxicity of Aβ or indirectly related to the impact of a lack of APOE on microglia/astroglia function remains to be determined.

## Discussion

Neuronal dysfunction and loss of synaptic integrity are tightly associated with aging and AD cognitive decline, but understanding the molecular bases of those changes and identifying possible modulators remain as important challenges to overcome. APOE, the major apolipoprotein of the brain and most important genetic contributor to the sporadic form of AD (Wisniewski & Frangione, 1992; Corder et al, 1993, 1994; West et al, 1994; Hyman et al, 1996; Lippa et al, 1997), is a well-established partner of amyloid β peptides and has been shown to directly impact the aggregation, deposition, and clearance of those neurotoxic species (Holtzman et al, 2000; Fagan et al, 2002; Deane et al, 2008; Castellano et al, 2011; Hashimoto et al, 2012; Koffie et al, 2012; Hudry et al, 2013). Although those findings clearly emphasize the pivotal role of APOE in AD neuropathological changes, the field is still divided as to whether or not clinical benefit could be achieved by reducing the levels of APOE (Koldamova et al, 2005; Bien-Ly et al, 2012; Liao et al, 2014; Lane-Donovan et al, 2016; Zheng et al, 2017). In addition, previous

studies have described that a complete lack of APOE in a physiological context may lead to a loss of brain function during aging. The question, therefore, is how can we disambiguate the function of APOE in maintaining brain resilience during aging while also participating in Aβ-induced neurotoxicity?

By recording visually evoked neuronal responses in awake mice, we were able to demonstrate that a complete absence of APOE restores neuronal function in adult APP/PSEN1 animals (decreased "off-responding" cells and increased OSI and DSI), so that the efficacy of induced responses is comparable between wild-type and APP/PSEN1/APOE[null] animals. In addition, the lack of APOE in AD mice preserves synaptic and neuritic integrity around plaques and decreases amyloid-associated glial immunoreactivity. Despite these beneficial effects, the idea of knocking-down *APOE* expression to improve AD neuropathological hallmarks remains controversial because previous reports have shown that APOE[null] animals present with some cognitive deficits, electrophysiological alterations, and functional connectivity changes when compared with wild-type mice (Masliah et al, 1997; Kitamura et al, 2004; Trommer et al, 2004; Yang et al, 2011a; Zerbi et al, 2014). Here, we confirm that genetic ablation of APOE impairs visually evoked responses of aged APOE[null] and APP/PSEN1/APOE[null] mice as compared with controls. Taken together, we demonstrate that *apoE* genetic disruption alleviates Aβ-associated neurotoxicity early but sensitizes the brain towards age-dependent neuronal dysfunction. Discrepancy between our results and previous reports may result from the experimental settings used, as we assessed neuronal function using a relatively simple sensory circuit within the brain as compared with more complex behavioral tasks.

Data in the human population are essentially missing, with only one case study published so far reporting "below-average" cognition and memory performances in an individual with a complete lack of APOE protein in the brain (Cullum & Weiner, 2015). Because of the relatively young age of the individual, it is hard to predict if

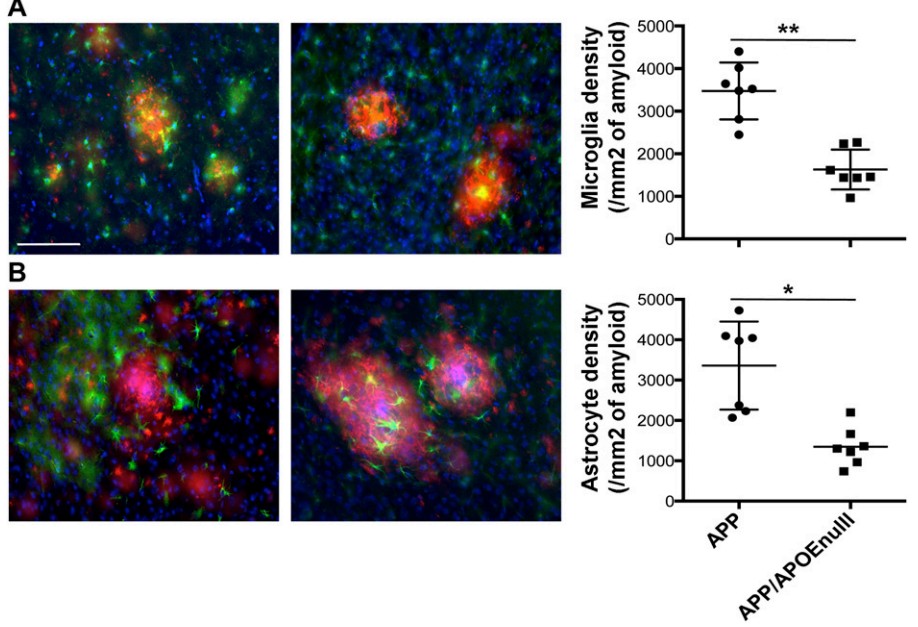

**Figure 6. Decreased glial reactivity in APP/PSEN1 mice lacking APOE.**
**(A)** RepreseAPP/PSEN1 and APP/PSEN1/APOE[null] adult mice. **(B)** Representative images (left panel) and stereological evaluation (right panel) of the density of GFAP-positive astrocytes (green) around amyloid deposits (red, after staining using a rabbit anti-Aβ antibody, IBL) in APP/PSEN1 and APP/PSEN1/APOE[null] adult mice. In each case, the total number of Iba-1–positive microglia or GFAP-positive astrocytes present in less than 50 μm around a plaque was reported to the surface of the plaque. Scale bar = 100 μm. n = 7 mice/group (10–12 mo "adult" cohort); Mann–Whitney test. *P < 0.005, **P < 0.001.

the absence of APOE will lead to exacerbated cognitive decline later on, similar to what is observed in animal models. Alternative strategies to inhibit the detrimental impact of APOE in disease while preserving its physiological role in maintaining brain homeostasis have already been attempted. For example, intraperitoneal injections of A$\beta$12-58P, a modified A$\beta$ peptide homologous to the binding side of APOE on A$\beta$40/A$\beta$42 peptides that crosses the blood–brain barrier, is able to disrupt the interaction between APOE and A$\beta$, reduce the formation of fibrillar aggregates, alleviate $\tau$ pathology in triple transgenic mice (PS1M146V, APPSwe, and tau-P30IL), and improve behavioral deficits (Yang et al, 2011b; Liu et al, 2014). Reducing the amount of APOE specifically in the brain tissue (and not in the periphery) could be another alternative. The recent characterization of the bEKO mouse model by Lane-Donovan and colleagues (Lane-Donovan et al, 2016), which fortuitously lacks APOE in the brain but shows normal levels in the plasma, suggests that this may be the case, as these animals do not show any cognitive deficits as compared with APOE[null] mice. However, the clinical translation of this discovery may prove difficult to achieve considering that various neural cell types express *APOE* (astrocytes, microglia, and endothelial cells). Interestingly, the work by Zheng and colleagues demonstrated that the sole deletion of APOE in astrocytes ameliorates the spatial learning and memory deficits of APP/PSEN1 mice (Zheng et al, 2017), eventually suggesting that targeting APOE in the "right cells" (assuming that APOE derived from astrocytes, microglia, or other cells within the brain has different biological properties and functions) may trigger clinical improvement.

Importantly, the "rescue effect" observed in the absence of APOE did not directly correlate with a significant change in the global amount of amyloid pathology in the brain of our adult cohort (assessed by conventional immunostaining or by quantification of A$\beta$ by ELISA), but rather with a specific decrease in dense-core Methoxy-positive deposits in APP/PSEN1/APOE[null] mice (both in adult and aged groups). Of note, the initial load of fibrillar amyloid at baseline is especially abundant in the model we used, the APP/PSEN1 strain, because of the presence of the Swedish APP (KM670/671NL) and $\delta$9 PS1-mutated transgenes. It is, therefore, conceivable that the impact of APOE deficiency may be less significant in other models of amyloidopathy that only carry a mutated APP allele. Nevertheless, the shift in the aggregation state of amyloid between APP/PSEN1 and APP/PSEN1/APOE[null] has been reported in other studies (Bales et al, 1997; Irizarry et al, 2000a) and essentially reproduces the morphological difference between $\beta$-pleated sheet deposits (labeled with Thioflavin-S) mostly observed in AD patients and diffuse plaques detected in non-demented individuals (Dickson, 1997; Urbanc et al, 2002). This conformational change of extracellular amyloid is relevant to the disease as fibrillary neuritic plaques are associated with deleterious effects on the surrounding neuropil, such as the presence of dystrophic neurites and recruitment of reactive glial cells (Masliah et al, 1990; Knowles et al, 1999; Vehmas et al, 2003), which are absent around more diffuse aggregates observed in cognitively normal individuals and in our APP/PSEN1/APOE[null] animals. The underlying mechanisms remain incompletely understood, but in vitro and in vivo studies have previously shown that APOE forms stable complexes with A$\beta$ (Naslund et al, 1995), directly co-deposits in plaques (Namba et al,

1991), modulates A$\beta$ oligomerization (Hashimoto et al, 2012; Garai et al, 2014), and converts amyloid protofibril to fibril (Hori et al, 2015), which all could result in the observed changes in plaque morphology. Further comparison of the profile of A$\beta$ oligomers, which are especially concentrated around amyloid plaques and have been shown to trigger neurotoxicity (Walsh & Selkoe, 2007; Arbel-Ornath et al, 2017; Yang et al, 2017), may give additional clues to understand the beneficial impact the lack of ApoE on A$\beta$-dependent neuronal function and synaptotoxicity.

It is possible that some of the protective effects we observed in APP/PSEN1/APOE[null] mice arise independently from the change observed in the aggregation state of A$\beta$. In particular, the decreased glial reactivity in APP/PSEN1/APOE[null] mice could result from a direct impact of APOE on the biology of microglia and astrocytes in the context of AD, even in the presence of A$\beta$ neurotoxic species. The APOE-TREM2 pathway has recently been identified as a major switch triggering a phenotypic change in microglia in the context of neurodegeneration (Atagi et al, 2015; Yeh et al, 2016; Krasemann et al, 2017). Upon activation of this cascade, microglial cells become chronically inflammatory and lose their homeostatic signature, a phenomenon that is not present when the endogenous expression of *apoe* was shut down specifically in microglia. Considering that activated neuroinflammatory microglia have also been shown to control the conversion of resting (A2) to reactive astrocytes (A1) (Liddelow et al, 2017), the inhibition of the TREM2-APOE pathway in APP/PSEN1/APOE[null] mice could explain the overall decrease of glial reactivity independently of the decreased density of neuritic plaques in those mice.

Finally, our array tomography data emphasize the importance of APOE in driving A$\beta$-associated synapse loss. While the density of synapsin and PSD95 puncta was dramatically reduced in the vicinity of amyloid deposits in APP/PSEN1 mice (by about 30%, as we have previously reported [Koffie et al, 2009; Kay et al, 2013]), no such reduction of pre- or post-synaptic elements was detected in APP/PSEN1/APOE[null] mice. Still, the lack of APOE neither prevented the recruitment of A$\beta$ at the synapse and nor the decreased synaptic volume characteristic of amyloidosis mouse models. As the synaptic size is generally accepted as an indicator of connection strength (Kasai et al, 2010; Penzes et al, 2011), it is possible that synaptic stability remains compromised in APP/PSEN1/APOE[null] mice. In addition, we have previously reported that the different APOE isoforms differentially modulate the recruitment of A$\beta$ at the synapse in human AD brains (Koffie et al, 2012), therefore acting as a chaperone toward neurotoxic amyloid peptides. The fact that A$\beta$ still co-localizes with synaptic terminals in the absence of APOE implies that other factors may be involved to shuttle those neurotoxic species at the synapse. Among others, apolipoprotein J (ApoJ, or clusterin) is a likely candidate, as it is also known to interact with A$\beta$ and modify its aggregation and deposition (Mulder et al, 2014; Miners et al, 2017). Whether or not this is the case remains to be determined, as we did observe a rescue of neuronal function in the absence of APOE alone, therefore indicating that the presence of A$\beta$ at the synapse is necessary but not sufficient to trigger synaptic collapse. Because our findings place APOE as a pivotal factor triggering amyloid synaptotoxicity, it is possible that its association with A$\beta$ is necessary for it to adopt a toxic conformation or allows the interaction between A$\beta$ and specific cellular receptors

(such as APP itself, as recently reported [Wang et al, 2017], TREM2 [Yeh et al, 2016], or any other APOE receptors [Holtzman et al, 2012; Lane-Donovan & Herz, 2017]) to initiate various deleterious molecular cascades in neurons or even in glial cells.

# Materials and Methods

### Animals

APPswe/PS1dE9 (APP/PSEN1) mice (The Jackson Laboratory [Jankowsky et al, 2004; Reiserer et al, 2007]) were used for the study. This mouse model expresses the human mutant APP and presenilin genes respectively containing the Swedish mutation K594N/M595L and the exon 9 deletion. Amyloid deposition starts as early as 3-mo old in this model, with most amyloid plaques being of fibrillar nature because of the presence of the $\Delta$E9 PS1 mutation leading to an increase in A$\beta$42/40 ratio. To generate APP/PSEN1/APOE$^{null}$ mice, one APP/PSEN1 hemizygous transgenic animal was crossed with an APOE$^{null}$ breeder (The Jackson Laboratory). The resulting APP/PSEN1/APOE$^{+/-}$ offspring was then crossed with another APOE$^{null}$ breeder to generate APP/PSEN1/APOE$^{null}$ and APOE$^{null}$ animals. The resulting APP/PSEN1/APOE$^{null}$ offspring was then bred with APOE$^{null}$ mice to generate all the animals used in this study. Controls included littermates from the APP/PSEN1 colony altogether with C57BL/6 (appropriate controls of the APOE$^{null}$ mice). We had previously verified that these control groups showed similar responses to visual stimulation and, therefore, could be merged together. Genotyping for APOE and APP were performed by PCR following the protocol given by The Jackson's Laboratory. A detailed description of the cohort is presented in Fig S1. All animal experiments were approved by the Massachusetts General Hospital Subcommittee on Research Animal Care following the guidelines set forth by the National Institutes of Health Guide for the Care and Use of Laboratory Animals.

### Viral vector construction and production

The yellow cameleon cDNA (YC3.6) was cloned to an AAV2 backbone, under a hybrid CMV immediate-early enhancer/chicken $\beta$-actin promoter/exon1/intron and before the woodchuck hepatitis virus posttranscriptional regulatory element (WPRE). High titers of AAV serotype 8 were produced using a conventional triple transfection approach by the PENN Vector Core (University of Pennsylvania). Viruses were tittered by quantitative PCR, and the final concentrations of the AAV viral stocks used for this study reached $4 \times 10^{12}$ vg/ml.

### Intracortical AAV injection and cranial window implantation

Surgical procedures were performed as described previously (Hudry et al, 2010; Kuchibhotla et al, 2014; Arbel-Ornath et al, 2017), with minor modifications. Mice were anesthetized by intraperitoneal injection of ketamine/xylazine (100 mg/kg and 50 mg/kg, respectively, body weight) and positioned on a stereotactic frame (Kopf Instruments). A 3-mm cranial window was opened

above the right primary visual cortex V1 (coordinates from $\lambda$: +0.5 mm anteroposterior, 2.7 mm lateral, and 0.6 mm dorsoventral). After removing the skull, 3 $\mu$l ($1.2 \times 10^{10}$ vg) of AAV8-CBA-YC3.6 viral suspension was injected at a rate of 0.15 $\mu$l/minute in the layer 2/3 neurons using a 33-gauge sharp needle attached to a 10-$\mu$l Hamilton syringe (Hamilton Medical). The window was then closed with a 5-mm-diameter cover glass and secured with a mixture of dental cement and Krazy Glue. A custom-made stainless steel headpost (Ponoko) was finally fixed to the skull using C&B Metabond dental cement (Parkell) to allow recording in awake mice. After a recovery period of one month, the mice were habituated to the head-fixation device (Thorlabs posts and Altos head clamps) by running freely on a circular treadmill (Ponoko) for 15 min a week before the first imaging.

### Recording of visually evoked neuronal responses

For in vivo calcium imaging, an Olympus FluoView FV1000MPE multiphoton laser scanning system mounted on an Olympus BX61WI microscope and an Olympus 25× dipping objective (NA = 1.05, Olympus) were used, with the emission path shielded from external light contamination. A DeepSee Mai Tai Ti:sapphire mode-locked laser (Mai Tai; Spectra-Physics) generated two-photon excitation at 860 nm, and detectors containing three photomultiplier tubes (Hamamatsu) collected emitted light in the range of 460–500, 520–560, and 575–630 nm. CFP and YFP photomultipliers (PMTs) settings remained unchanged throughout the different imaging sessions, but laser power was adjusted as needed. The mice were imaged awake and directly placed under the objective. To detect visually evoked neuronal responses, time courses of calcium transients were recorded in head-fixed awake animals while a 19-inch LCD monitor (Viewsonic VP930B) displayed the visual stimuli in front of the left eye (screen–eye distance, ~20 cm; screen-midline angle, 60°). Visual stimuli were drifting (2 Hz) sine-wave gratings (80 or 100% contrast, black and white) presented for 7 s. Eight stimuli (at 45° orientation increments) were presented sequentially in counter-clockwise order with a 7-s pause between stimuli, and 10 cycles of visual stimulation were presented during each recording. For each imaging session, two to three cortical frames (254 $\mu$m × 254 $\mu$m, scan rate 2.3 Hz, 0.429 s/frame, depth of 200–300 $\mu$m) were taken.

### Image processing and analysis

In vivo imaging data were analyzed using custom-written scripts in Fiji (National Institutes of Health: http://fiji.sc/) and MATLAB (MathWorks). Images were aligned for shifts in the x–y plane using the StackReg function of Fiji, neuronal cell bodies (identified by the presence of neuritic processes) were manually selected (ROIs), YFP: CFP ratios were created and spatially filtered, and raw time courses were extracted. "Responsive" cells were detected by comparing the change in directional response ($\Delta$R; R – Rnonstimulated) across directions and against background, and ($P < 0.05$, ANOVA with the Tukey–Cramer post hoc test and Bonferonni correction). Only responsive cells were used for further analysis of orientation and direction tuning. The OSI and DSI were calculated as described previously (Kuchibhotla et al, 2014). In short, OSI was calculated as the maximum directional response, $\Delta$Rmax, and divided by the sum

of responses in all other directions, ΣΔRother (0 ≤ OSI ≤ 1; 1 = perfectly orientation-tuned). DSI was calculated as ΔRmax divided by the sum of that direction and the antiparallel direction, ΔRmaxr+ ΔRorthogonal (0 ≤ DSI ≤ 1; 1 = perfectly direction-tuned). The higher the values for OSI and DSI, the better the neuron tuned to a specific visual stimulus.

## Tissue collection and processing

Mice were euthanized by $CO_2$ asphyxiation and tissue collected for immunohistochemical and biochemical analysis. For the mice that underwent in vivo imaging, the right hemisphere (containing the surgical site) was fixed by immersion in 4% paraformaldehyde and 15% glycerol in PBS for 48 h before cryoprotection with 30% glycerol in PBS. A few 1-mm$^3$ pieces of tissue were cut across the left hemisphere for array tomography. Because the surgical procedure and the injection of AAV-CBA-YC could have compromised the results of our stereological analyses, another cohort of APP/PSEN1 and APP/PSEN1/APOE$^{null}$ mice was used to perform the stereo-logical work (right hemisphere) and the isolation of synapto-neurosomes (from the left hemisphere directly snap-frozen in liquid nitrogen and stored at −80°C). Additional mice were euthanized to perform the sequential brain extraction from cortices.

## Synaptoneurosomes preparation and Western blotting

Synaptoneurosome preparations were based on procedures de-scribed by Tai et al (2012) with minor modifications. Briefly, an entire mouse hemisphere (about 250 mg of frozen tissue, without cere-bellum and olfactory bulbs) was homogenized in 1.2-ml cold Buffer A (25 mM Tris, pH 7.5, 120 mM NaCl, 5 mM KCl, 1 mM $MgCl_2$, 2 mM $CaCl_2$, 1 mM dithiothreitol, and Complete protease inhibitors; Roche), before being filtered through 80-$\mu$m nylon filters (Millipore). An aliquot of the filtrate was supplemented with SDS to 1%, boiled for 5 min, and centrifuged at 15,000 $g$ for 15 min, and the supernatant was collected as total extract. The other portion was further filtered through 5-$\mu$m pore filters (PALL Acrodisc) and centrifuged at 1,000 $g$ for 10 min at 4°C to pellet synaptoneurosomes. The supernatant, corresponding to the cytosolic extract, was further centrifuged at 100,000 $g$ for 20 min to remove microsomes or other extracellular vesicles. The synaptoneurosome pellet was washed once with cold Buffer A and centrifuged again at 1,000 $g$ for 10 min. The pellet was extracted with 0.5 ml Buffer B (50 mM Tris, pH 7.5, 1% SDS, and 2 mM dithiothreitol) and boiled for 5 min. After centrifugation at 15,000 $g$ for 15 min, the supernatant was collected as synaptoneurosomal extract. Protein content was measured by BCA assay (Pierce). The protein concentration was adjusted to 10 $\mu$g/10 $\mu$l for all Western blot samples, including Laemmli blue buffer and reducing agent. The samples were electrophoresed on 4–12% Bis-Tris gels in MES running buffer (Invitrogen). After transferring on nitrocellulose membrane (GE Healthcare), the blots were blocked for 1 h at RT in Odyssey Blocking Buffer (Li-Cor), before probing the membranes overnight at 4°C with the primary antibodies diluted in blocking buffer (1:1,000, Table 1). Incubation with IRDye 800CW or IRDye 680CW secondary antibodies (Li-Cor) followed (1:2,000) before detection of the near infrared signals using an Odissey CLx imager system (Li-Cor). Signal intensity was measured by densitometry using the Integrated Density function of Fiji software (http://fiji.sc/) and divided to GAPDH signal intensity. The samples from each batch of SNS were first normalized to the signal observed in the controls (Littermates), before performing statistical analyses on all SNS preparations.

## Sequential brain extraction

Sequential brain extraction was performed as described previously (Hashimoto et al, 2012) from one cerebral hemisphere, after dis-section of the olfactory bulb and cerebellum. The tissue was initially homogenized in 10 volumes of TBSI (Tris-buffered saline with pro-tease inhibitor cocktail; Roche) with 25 strokes on a mechanical Dounce homogenizer and centrifuged at 100,000 $g$ for 30 min at 4°C. The supernatant was collected as TBS-soluble fraction. The resulting pellet was consecutively extracted with TBS buffers containing 2% Triton and 2% SDS (with protease inhibitors), alternating homoge-nization and ultracentrifugation steps (100,000 $g$ for 30 min). The final pellet was solubilized in 500 $\mu$l of FA by sonication. After a final ultracentrifugation step, the FA-soluble fraction was desiccated and the pellet resuspended in 100 $\mu$l of DMSO. The content in the most soluble (TBS) and insoluble (FA) A$\beta$ species was analyzed in this study. The concentrations of A$\beta$40, A$\beta$42, and A$\beta$ oligomers were respectively quantified with a mouse/human ELISA kit (Wako) and a human Amyloid $\beta$ oligomers (82E1-specific) assay (IBL international).

## Immunohistology

After cryoprotection of the brain in 30% glycerol for 48 h, 40-$\mu$m-thick floating sections were cut on a freezing microtome in the sagittal plane. Floating sections were successfully permeabilized in 0.5% Triton in TBS for 30 min, blocked in 5% normal goat serum in TBS for 1 h, and incubated with primary antibody overnight at 4°C in 2.5% NGS and 0.1% Triton in TBS (see Table 1 for a complete list of primary antibodies used). The sections were then washed with TBS and in-cubated with appropriate Alexa-Fluor 488– or Alexa-Fluor 568–conjugated secondary antibodies diluted in 2.5% NGS and 0.1% TritonX in TBS. After another round of washing, the sections were mounted onto slides and coverslipped with VECTASHIELD Mounting Medium with DAPI (Vector Labs). For the counterstaining of amyloid with Methoxy-XO$_4$, the floating sections were incubated for 15 min in a solution of 1 $\mu$g/ml of Methoxy-XO$_4$ (diluted in TBS) before mounting the slices with Fluoromount-G (No DAPI, SouthernBiotech).

## Stereology-based quantitative analyses

All pathology quantification was carried out blinded until the last statistical analyses. Stereology-based studies of amyloid-associated neuritic dystrophies, reactive astrocytes, and microglia were performed on immunolabeled sections using an Olympus BX52 epifluorescent microscope equipped with motorized stage, DP70 digital CCD camera, and CAST stereology software (Olympus). The cortex was outlined under low-power objective (4×), and dystrophies, astrocytes, and microglia counts were made using 20× high numerical aperture (1.2) objective. Using a meander sampling of 70% of cortical area, images were captured each time an amyloid deposit was encountered. Those images were then analyzed using

**Table 1. List of antibodies used for the study.**

| Antigen | Species | Polyclonal/Monoclonal | Manufacturer | Cat. No |
|---|---|---|---|---|
| Antibodies used for Immunohistology (IH)/WB | | | | |
| Aβ (N-terminal) | Rabbit | Polyclonal | IBL/Tecan | 18584 |
| Aβ 1-12 (clone BAM10) | Mouse | Monoclonal | Sigma-Aldrich | A3981 |
| Iba1 | Rabbit | Polyclonal | Wako | 019-19741 |
| GFAP | Mouse | Monoclonal | Sigma-Aldrich | G3893 |
| Neurofilament (SMI-311R) | Mouse | Monoclonal, IgM, IgG1 | BioLegend | 837801 |
| GABA B receptor 1 | Mouse | Monoclonal, IgG 2a | Abcam | ab55051 |
| NMDAR 1 | Rabbit | Monoclonal | Abcam | ab109182 |
| NMDAR2B | Rabbit | Polyclonal | Abcam | ab65783 |
| NMDAR2A | Rabbit | Monocloncal, IgG | Millipore | 04-901 |
| GABA A receptor α1 | Rabbit | Polyclonal | Abcam | ab33299 |
| PSD95 | Rabbit | Monoclonal | Cell Signaling | 3450S |
| GAPDH | Mouse | Monoclonal | Abcam | ab9484 |
| Actin | Mouse | Monoclonal | Sigma-Aldrich | A4700 |
| Antibodies used for array tomography | | | | |
| Oligomeric Aβ (1C22) | Mouse | Monoclonal | Dr. Walsh's lab | Yang, O'Malley et al (2015) |
| Synapsin 1 | Rabbit | Polyclonal | Millipore | AB1543P |
| PSD95 | Guinea-pig | Polyclonal | Synaptic Systems | 124014 |

Fiji, counting the number of GFAP-positive astrocytes, Iba1-positive microglial cells or neuritic dystrophies (visible after immunostaining for neurofilaments) close to a plaque (<50 $\mu$m) and reporting this number to the surface of the plaque considered. For the quantification of amyloid load and amyloid density, Alexa-568-anti-Amyloid and Methoxy-XO4–positive plaques were imaged using a NanoZoomer-XR digital slide scanner (Hamamatsu) under a ×20 objective. After conversion of the .ndpis files to tiff format, the number of deposits and their surface were determined using a custom-written script based on the "Analyze particle" function of Fiji (National Institutes of Health: http://fiji.sc/). The total surface occupied by amyloid and the total number of plaques were then reported to the cortical area of each section considered.

### Array tomography

#### Sample collection and tissue processing
Array tomography analyses were performed as previously described (Hudry et al, 2013; Kay et al, 2013). Five to six pieces of cortical tissue (1 mm$^3$) were dissected and fixed for 3 h in 4% paraformaldehyde and 2.5% sucrose in 0.01 M PBS. After dehydration in ethanol, the samples were incubated in LR White resin (Electron Microscopy Sciences) overnight at 4°C before polymerization at 53°C. Ribbons of 20–40 ultrathin (70 nm thick) serial sections were cut with a Histo Jumbo diamond knife (Diatome) on an ultracut microtome (Leica) and mounted on glass coverslips.

#### Staining and imaging
Ribbons were incubated in glycine (50 mM glycine in 1× TBS) for 5 min and blocked in blocking solution (0.05% Tween and 0.1% fish gelatin [Sigma-Aldrich] in 1× TBS) for 1 h before antibody staining. All antibodies (summarized in Table 1) were spun for 4 min at 10,000 g before being applied. Mouse 1C22 against oligomeric Aβ (1:200, donated by Dominic Walsh [Yang et al, 2015], rabbit anti-synapsin-1 (1:50, Rb X Synapsin I, AB1543P; Millipore) and guinea pig anti-PSD95 (1:50, Anti-PSD95, 124014; Synaptic Systems [Koffie et al, 2009]) primary antibodies in block solution were added and incubated at 4°C overnight. Incubation with secondary antibodies diluted 1:50 in block solution (Alexa FluorTM 488 donkey anti-mouse IgG [H+L], Invitrogen; Alexa FluorTM 594 donkey anti-rabbit IgG [H+L], Invitrogen; and Alexa FluorTM 647 goat anti-guinea pig IgG [H+L], Invitrogen) was performed for 1 h at room temperature, before counterstaining with DAPI for 5 min. Shandon Immu-Mount (Thermo Fisher Scientific) solution was used to mount the slides on polysine microscope slides (VWR International). Serial sections of the ribbon were imaged with a Zeiss AxioImager Z2 epifluorescent microscope, first at 10× to obtain a tilescan of the entire ribbon, then with a 63 × 1.4 NA Plan Apochromat objective for high-resolution images. Images were acquired with a CoolSnap digital camera and AxioImager software with array tomography macros (Carl Zeiss, Ltd). In short, two distinct areas were selected on each 10× tilescan. Once the areas were selected on two serial slices of the ribbon, the AxioImager software was able to find the same areas in each one of the subsequent slices and image them at 63×.

#### Data analyses
Image stacks were aligned using ImageJ (National Institutes of Health open software; multistackreg macro [Thevenaz et al, 1998]). ROI (10 $\mu$m × 10 $\mu$m) were cropped on the stack near plaques (<10 $\mu$m from the plaque edge) and far away from plaques (>35 $\mu$m from the edge of the halo). When no plaques were present (in wild-type and APOE$^{null}$

mice), ROI were randomly selected on the stack. Custom algorithms were used to threshold the crops in IMAGEJ/FIJI (Schindelin et al, 2012). Custom MATLAB macros were used to remove puncta that were only found in a single section, detect synapses, quantify the numbers and sizes of synaptic puncta, and determine which synaptic puncta were co-localized with Aβ. Synaptic density was calculated as the number of puncta per volume of tissue sampled (synapses/mm$^3$).

### Statistical analyses

A detailed summary of the statistical analyses performed is presented in the Supplemental Information 1 "Statistical analyses," including the test chosen for each analysis, exact *P*-values and confidence intervals. The statistical software SAS was used to analyze the relationship between adjusted DSI/OSI and genotype, age, and the interaction between them. A linear mixed model was fitted to the DSI and OSI values after log transformation, with these factors (genotype or age) as fixed effects and mouse as random effect. Significance was set for *P*-values < 0.05.

Array tomography data were analyzed using GraphPad Prism software, and $P < 0.05$ was considered significant. Shapiro–Wilk test was used to check for normal distribution of data. Brown–Forsythe test (with ANOVA or the nonparametric equivalent Kruskal–Wallis test) and F-test (with *t* test or the nonparametric equivalent Mann–Whitney test) were used to check for equality of variance of the data. Synaptic density in crops was measured and averaged for each mouse. A two-way ANOVA test followed by a Tukey's multiple comparisons post hoc test was used to compare synaptic density using distance from plaques and mouse genotype as the two variables. A *t* test was used to compare the percentage of 1C22 co-localizing at synapses close to plaques in APP and APP/APOE$^{null}$ mice. One-way ANOVA test (or the nonparametric equivalent Kruskal–Wallis test) followed by a Holm Sidak's or a Tukey's multiple comparisons post hoc tests and *t* test (or the nonparametric equivalent Mann–Whitney test) were used to compare synaptic volume across genotypes far and close from plaques.

All the other results (stereology, Western blotting) were analyzed using GraphPad software. Normality was initially validated using the D'Agostino and Pearson omnibus normality test. Difference between each group was tested using a one-way ANOVA (in case of normality) or a Kruskal–Wallis test (non-normal distribution) followed by with post hoc Dunn's multiple comparison test. In case of comparisons between two groups only (APP/PSEN1 and APP/PSEN1/APOE$^{null}$), a conventional *t* test (normality) or a Mann–Whitney test (non-normal distribution) was performed. Data are presented as mean ± SD.

# Supplementary Information

# Acknowledgements

This work was supported by the National Institute of Health/the National Institute on Aging 1K99AG047336-01A1 (E Hudry) and 5R01AG047644-04
(BT Hyman). T Spires-Jones, R Jackson, and C Cannavo are supported by the UK Dementia Research Institute, European Research Council, Alzheimer's Research UK (ARUK-SPG2013-1), Wellcome Trust-University of Edinburgh Institutional Strategic Support Fund, and Alzheimer's Society (AS-PG-15b-023). T Spires-Jones is a member of the Federation of European Neuroscience Kavli Network of Excellence.

## Author Contributions

E Hudry: conceptualization, resources, data curation, formal analysis, supervision, funding acquisition, validation, investigation, visualization, methodology, project administration, and writing—original draft, review, and editing.
J Klickstein: data curation, formal analysis, and writing—review and editing.
C Cannavo: data curation, formal analysis, and writing—original draft, review, and editing.
R Jackson: data curation, formal analysis, methodology, and writing—original draft, review, and editing.
A Muzikansky: formal analysis, methodology, and writing—original draft, review, and editing.
S Gandhi: data curation, formal analysis, and methodology.
D Urick: data curation and formal analysis.
T Sargent: data curation and formal analysis.
L Wrobleski: data curation and formal analysis.
AD Roe: data curation, formal analysis, and project administration.
SS Hou: formal analysis and methodology.
KV Kuchibhotla: supervision, methodology, and writing—original draft, review, and editing.
RA Betensky: formal analysis, supervision, methodology, and writing—original draft, review, and editing.
T Spires-Jones: conceptualization, supervision, funding acquisition, methodology, and writing—original draft, review, and editing.
BT Hyman: conceptualization, formal analysis, supervision, funding acquisition, and writing—original draft, review, and editing.

## Conflict of Interest Statement

The authors declare that they have no conflict of interest.

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
