## [Reviewer comments · Life Science Alliance]

Life Science Alliance

OPPOSING ROLES OF APOLIPOPROTEIN E IN AGING AND NEURODEGENERATION

Eloise Hudry, Jacob Klickstein, Claudia Cannavo, Rosemary Jackson, Alona Muzikansky, Sheetal Gandhi, David Urick, Taylie Sargent, Lauren Wroblewski, Allyson Roe, Steven Hou, Kishore Kuchibhotla, Rebecca Betensky, Tara Spires-Jones, and Bradley Hyman

DOI: <https://doi.org/10.26508/lsa.201900325>

Corresponding author(s): Eloise Hudry, Massachusetts General Hospital - Harvard

Review Timeline:

Submission Date:	2019-01-29
Editorial Decision:	2019-01-30
Revision Received:	2019-02-05
Accepted:	2019-02-06

Scientific Editor: Andrea Leibfried

Transaction Report:

Please note that the manuscript was previously reviewed at another journal and the reports were taken into account in inviting a revision for publication at *Life Science Alliance* prior to submission to *Life Science Alliance*.

Referee #1:

This manuscript was designed to determine the effect of endogenous mouse (m)-apoE on aging in wildtype (wt) mice and AD-related pathology in APP/PS1 mice. Thus, the comparison is wt vs. APOE^{-/-} vs. APP/PS1 vs. APP/PS1/APOE^{-/-}. The readouts include visually-evoked calcium transients, amyloid deposition, aggregation state of A β , levels of extracted A β 40 and A β 42, levels of excitatory or inhibitory neurotransmitter receptors at the synapse (including adjacent to plaques), neuroinflammation (astrocytosis and microgliosis). The manuscript is carefully written and the experiments are well designed and executed. The major problem is conceptual.

1. In wt mice, removing endogenous m-APOE has primarily a negative effect, including "lack of APOE independently triggers mild neuronal dysfunction with age". This makes intuitive sense as m-apoE is interacting with the endogenous expression levels of other mouse proteins.

Answer: We agree with this statement, which emphasizes on the important role of apoE in the maintenance of neuronal homeostasis and function (as further demonstrated in our study).

2. In contrast to wt mice, in FAD-transgenic (Tg) mice, m-apoE is interacting with over-expression of human (h)-APP/PS1 and the resulting h-A β 40 and A β 42. As far back as Bales in 1997, knock-out of m-APOE from PDAPP mice improved the A β pathology of that FAD-Tg mouse model. Most recently, Ulrich, JEM, 2018 reported that "ApoE facilitates the microglial response to amyloid plaque pathology", but this paper was, again, describing the effects of removing m-APOE from a FAD-Tg mouse model. Overall, that is the same result reported herein. "We concluded that m-APOE is a necessary factor participating in amyloid-dependent loss of network integrity". The assumption remains that m-apoE, which differs from h-apoE by ~100/299 amino acids, has an essentially toxic interaction with h-APP/PS1 and the resulting h-A β 40 and A β 42 and the pathology they induce. However, when h-APOE is expressed in FAD-Tg/APOE^{-/-}, even the expression of APOE4 improves the pathology compared to APOE^{-/-}. For this reason, experiments designed to determine the difference between the presence and absence of m-apoE would seem not as relevant as the difference between the h-apoE isoforms (with a single amino acid difference) on a FAD-Tg mouse background. Indeed, experimental FAD/hAPOE models have produced data relevant to the differences in AD risk caused by APOE4 and APOE2.

Answer: We are well aware of the limitations of using mouse models to mimic complex neurological diseases and of the value of comparing the impact of the different human APOE isoforms as compared with murine apoE. However, the present study was essentially designed to get a better understanding of the fundamental impact of the apoE protein on neuronal function. This work was not primarily designed to address how apoE affect amyloid pathology or microglial phenotype, as those aspects have already been extensively investigated (as stated by the reviewer). Rather, knowing the many functions of apoE in the brain, we aimed to assess how its absence would drive changes in the ability of neurons to integrate and respond to sensory stimuli. This work does not provide further information about which specific aspect of apoE biology (on lipid metabolism, neuroinflammation, blood brain barrier integrity, etc) drives the

rescue of neuronal dysfunction in the context of AD Tg mice or leads to neuronal dysfunction in aged APOE-null mice in absence of amyloid pathology.

3. Even more concerning is the conclusion that "down-regulating ApoE expression as a possible therapeutic approach to alleviate A β -dependent neuronal impairment." Perhaps this would be an appropriate therapeutic approach for a FAD-Tg mouse. But without the interactions between h-APP/PS1 and the h-apoE isoforms, this is a potentially dangerous conclusion.

Answer: We respectfully disagree with this statement. Indeed, our study shows that the absence of apoE improves Abeta-related neuronal dysfunction and synaptic loss, which clearly establish apoE as an important partner of Abeta to induce neurotoxicity. Based on those readouts, hypothesizing that down-regulating apoE expression in patients could be a way to alleviate Abeta-dependent neurotoxic effects and be beneficial to AD patients makes sense. A recent paper published in 2017 by the group of Dr. Holtzman at WashU has attempted such therapeutic approach in APOE4 AD Tg mice using antisense oligonucleotides targeting apoE (PMID 29216448). This work showed that a significant decrease of the levels of apoE mRNA and protein could be achieved after intracerebroventricular injection of ASO, and that lowering apoE after A β seeding modulates plaque size and toxicity (similarly as the present study).

Referee #2:

In this manuscript, the authors try to address the question whether loss of Apoe had any detrimental consequences (in wild type mice) and can reduce Abeta amyloid damage (in APP/PS1 mice). The authors use an experimental paradigm based on measuring neuronal activation in the visual cortex. They show that "the absence of APOE largely abrogates A β -dependent neuronal dysfunction in a mouse model of amyloidosis" and that the "complete lack of APOE itself can impact tuning performances". The data support well their conclusions, although the effects seen are not high. Further, they show that lack of Apoe is beneficial at the level of amyloid-induced synaptic damage (loss of Apoe rescues the decrease in synaptic density). A very interesting finding is in the last part of the paper, where the authors show that lack of Apoe abrogates astrogliosis and microgliosis around amyloid plaques. The study does not conclusively address whether Apoe loss is a viable therapy, but is an interesting report nonetheless adding to the knowledge of this field.

Remarks:

The authors should provide information about power calculations for the number of mice that they have decided to use.

Answer: We have not performed a power calculation beforehand for the present work but evaluated the number of animals needed based on a previous work from our lab using a similar experimental setting in a model of tauopathy (in vivo recording of visual-evoked neuronal responses in wild-type and Tg4510 mice, Kuchibhotla KV et al. PNAS. PMID 24368848). Another paper published in 2012 reported visual-dependent decline of neuronal function in a

mouse model of amyloidosis with comparable features as the one used in the present work (Grienberger C et al. Nat Comm. 2012, PMID: 22491322). The results of those seminal studies reached significance with a number of mice of 3-15 mice/group, as compared with 5 <n <11 animals per group in our work.

Authors should refrain from using "trend" when a comparison does not show a p value below their set threshold (e.g. page 4 when commenting Figure EV3, or page 5 last paragraph).

Answer: We have now corrected this issue and disambiguate those statements for more clarity.

On page 5, first line, authors say that "non-responder" neurons were almost absent in all groups (except APP/PS1-Apoenull), however looking at Figure 2C there seem to be cells in all groups, particularly in Apoenull group. Authors should reword this sentence, and possibly have numbers of percentages of non-responders for each group in the text, as it is difficult to extrapolate from the plot in 2C.

Answer: We have now rephrased this sentence and included the percentages of "off-responders" for each experimental group:

"Intriguingly, the percentage of off-responding neurons was much lower in all the other groups considered (wild-type: $4.32 \pm 1.27\%$; APOEnull: $4.68 \pm 1.19\%$; APP/PSEN1/APOEnull: $3.2 \pm 0.84\%$), and especially in APP/PSEN1/APOEnull mice lacking apoe expression, suggesting that APOEnull mice normalizes an alteration from normal physiology observed in APP/PSEN1 mice."

Loss of Apoe causes a drastic decrease in the number of microglia and astrocytes recruited at plaques. Could the authors add a quantification of microglia and astrocytes in wild type and wild type-Apoenull? It would be quite interesting to add the information about whether loss of Apoe in APP/PS1 mice results in a completely abolished recruitment (i.e. there is no difference with a general astrocyte / microglia distribution in the parenchyma of wild type mice, meaning that in absence of Apoe plaques are gliosis-inert).

Answer: The gross evaluation of the density of microglial cells between wild-type and apoE-null mice did not reveal any striking difference, and therefore we did not perform a complete stereological evaluation to confirm those findings.

Referee #3:

Hudry et al. investigated apolipoprotein E (ApoE) effects on neuronal function and synaptic integrity in adult wild-type, Apoenull, APP/PS1 and APP/PS1;Apoenull mice. Using two-photon calcium imaging to record visually-evoked responses, authors first found that genetic removal of Apoe improved neuronal responses in 8-10 months of age APP/PS1 mice. In addition, these mice developed fewer parenchymal amyloid plaques with a reduced amount of synaptic loss - using high-resolution array tomography - and activated glial cells. Secondly, they also demonstrated that neuronal function is disrupted in aged mice (18-20 months of age) lacking Apoe, even in absence of amyloid. They concluded that ApoE has a dual effect with a neurotoxic component

during early stages of amyloid brain load and a neuroprotective component in later stages of aging. The effect of lack of murine ApoE in models of amyloidosis has been extensively studied for the past two decades. Several concerns exist about the current study, as described below:

1. Several studies have shown that ApoE^{-/-} mice exhibit a vascular phenotype including pronounced blood-brain barrier (BBB) leakage of blood-derived molecules, increased BBB permeability to Gadolinium contrast agent, perivascular IgG and fibrinogen deposits, loss of tight junctions, increased MMP-9 cerebrovascular expression, loss of pericytes, basement membrane degeneration, etc. (see for example Am. J. Mol. Med. 2001; 7:810-815; Am. J. Physiol. Cell Physiol. 2007; 292:C1256-C1262; Exp Neurol. 2001 May;169(1):13-22; Lab Invest. 2001 Jul;81(7):953-60; J Biol Chem. 2011 May 20;286(20):17536-42; Nature. 2012 May 16;485(7399):512-6; PLoS Biol. 2015 Oct 29;13(10):e1002279; Mol Psychiatry. 2014 Oct;19(10):1143-9; Ann Neurol. 2016 Jan;79(1):144-51; Front Physiol. 2016 Oct 6;7:453). Did the authors observe an effect on cerebral vasculature after genetic ablation of ApoE? Did ApoE loss cause vascular dysfunction in aged mice that could mediate the observed neuronal dysfunction and visually-evoked responses?

Answer: We did not focus our analysis on the impact of the lack of apoE on the integrity of the neurovasculature, as we primarily analyzed its effect on neuronal function. We agree with the reviewer that apoE has been implicated in the maintenance of the BBB integrity, a function that could definitively contribute to the observed phenotype. Other important roles of apoE in brain lipid metabolism, myelin integrity, neuroinflammation, neuronal response to stress or via its direct interaction with Abeta could participate in apoE-dependent preservation of neuronal function in APP/PS1 mice or loss of neuronal functionality in aged apoE-null mice. The relationship between the vascular anomalies caused by the lack of ApoE and their effect on neuronal and synaptic integrity could potentially be the object of another study.

2. Plasma cholesterol is typically high in ApoE^{-/-} mice. Has the same been observed after global ApoE silencing? Would silencing ApoE in the central nervous system have different effect from global ApoE silencing?

Answer: We apologize if our manuscript was not clear enough and the description of the mouse model used in our study confusing. We have actually used mice presenting with a complete knock-down of the murine *apoE* gene in both the nervous system and the periphery. Unfortunately, no mouse model yet allows to conditionally ablate the *apoE* gene in the nervous system or the liver independently. We agree that it would be a very powerful approach to be able to do so.

3. How does ApoE genetic ablation rescue synaptic integrity near Abeta plaques? Is this a direct or indirect effect? Investigating some mechanistic aspect would greatly strengthen the present findings.

Answer: We agree with the reviewer that this is a very interesting finding, especially because previous papers have demonstrated that ApoE may act as a chaperone for the recruitment of Abeta neurotoxic species at the synapse (Koffie RM et al. Brain 2012). Intriguingly, our array tomography data do not show that there is necessarily less Abeta at the synapse, which goes

against this initial hypothesis. In our manuscript, we therefore discuss the possibility that the rescue effect toward Abeta-dependent synaptic loss observed in absence of apoE may be due to the fact that the association between Abeta and apoE changes the conformational state of those neurotoxic species and leads to more harmful impact on synapses. In vitro, apoE has been previously shown modulates the formation of neurotoxic Abeta oligomers (Hashimoto T et al. J. Neurosci. 2012), but we have not assessed this phenomenon in the present study.

4. The authors findings show "that the absence of APOE largely abrogates A β -dependent neuronal dysfunction in a mouse model of amyloidosis." Other groups have shown that ApoE^{-/-} mice have a phenotype with cognitive behavioral deficits and neuronal dysfunction (Nature. 2012 May 16;485(7399):512-6; Front Aging Neurosci. 2016 Nov 29;8:287; Behav Sci (Basel). 2018 Mar 3;8(3)). In the context of ApoE deficiency, how is neuronal function impaired in the absence of Abeta (refs above) and neuronal function improved in the presence of Abeta (authors' findings)? Please expand on this point and include a working model of how ApoE ablation improves specifically Abeta-dependent neuronal dysfunctions.

Answer: The reported cognitive deficits in apoE-deficient animals remain somewhat controversial, as stated in the introduction of our manuscript (a few papers demonstrated cognitive impairment in those mice while others failed to do so: Grootendorst et al., 2008, Hartman, Wozniak et al., 2001). This is exactly why we initiated the present work, so that we could get a better understanding of the impact of the lack of apoE on neuronal function at the level of single cells (using multiphoton calcium imaging) and within a simple neuronal circuit (the visual network). Our findings do not disagree with the above references highlighted by the reviewer. Indeed, we also detected some impairment in the visually-evoked neuronal responses in aged ApoE-null mice, even without any amyloid pathology. In the discussion of our manuscript, we emphasize the possible molecular mechanisms that could explain why the lack of apoE alleviates Abeta-dependent neurotoxicity while at the same time sensitizing the brain to stress in older animals, even without amyloid pathology.

5. The rationale to examine neuronal function and synaptic integrity specifically in the visual cortex should be provided. Does neuronal dysfunction occur in other brain regions relevant to AD pathology?

Answer: The study of neuronal dysfunction in the visual cortex is relevant to AD, as deficits in central sensory processing have been reported in the disease, particularly at advanced stages (PMID: 19713001, 1996878). In addition, the visual cortex is a convenient model to assess network integrity in vivo because it is easily accessible to perform either electrophysiology recordings or intra-vital calcium imaging (PMID: 22196337). The visual area V1 is also the output of a relatively simple circuitry, downstream of the retina and the lateral geniculate nucleus within the thalamus (PMID: 28772103), therefore facilitating the study of neuronal integration and response to well-controlled sensory stimuli.

We have now modified our manuscript and added this justification for our approach in the first paragraph of the results.

6. Fig.4A: The overall level of Abeta doesn't change in APP/PS1 vs. APP/PS1;Apoenull mice, however the distribution appears different between the groups (larger in diameter/size in

APP/PS1;Apoenull mice). How does this impact neuronal function in APP/PS1;Apoenull mice? One would assume that neuronal damage would be greater in areas with high Abeta intensity. Authors should take this into account and re-analyze their current data in relation to neuronal function and synaptic integrity along with glial activation.

Answer: We have demonstrated in the present study that, even though the global load of amyloid is comparable in APP/PS1 and APP/PS1/apoE-null mice, the amyloid deposits observed in APP/PS1/apoe-null mice are mostly “diffuse and non-neuritic”, similarly to the deposits observed in non-demented aged individuals. Because of this striking change in the aggregation state of Abeta peptides, it is not very surprising that no deficit in the visually-evoked neuronal responses are observed in AD Tg mice lacking apoE. By contrast, the plaques detected in AD transgenic mice expressing apoE are dense (labelled by Methoxy-XO4) and potentially much more neurotoxic, which is why pronounced deficits are observed in those mice. In our manuscript, we extensively discussed the fact that there is no strong correlation between the overall amyloid burden and Abeta-dependent neurotoxicity. Rather, the state of Abeta aggregation matters.

7. Does ApoE exert differential effects on neuronal numbers during early neurotoxic stage (e.g. loss) and later neuroprotective stage in the aging brain?

Answer: We did not perform an extensive evaluation of the density of neurons in the brains of apoE-null or APP/PS1/apoE-null mice, but we did not notice any sign of neuronal loss after gross evaluation of the cortex of those mouse models (evaluation of the cortical thickness or of the intensity of NeuN immunostaining). The initial characterization of the APP/PS1 mouse model also did not report any overt neuronal loss (PMID 14645205).

8. What is the efficiency of ApoE knockdown in the present model as measured for example by apoE levels in plasma and CSF?

Answer: The *apoE* knockdown mouse model used in this study (available at the Jackson Laboratory) has been initially developed by disrupting the endogenous mouse *apoE* gene in embryonic stem cells. This mouse line is homozygous for this genetic modification and therefore does not express any detectable apoE mRNA or protein across the entire organism. As such, apoE levels in plasma and CSF are not detectable.

9. Figures 1-3 do not indicate whether adult (8-10 month-old) or aged (18-20 month-old) mice were studied. These details should be indicated in the figure legends.

Answer: We have now added this information in the figure legends. The results presented in figures 1 to 3 correspond to data from adult and aged mice altogether.

Minor Comments:

10. What is reference [1] cited in the first sentence of the Introduction? The reference list does not include references cited by numbers.

Answer: We apologize for this error and have now inserted the correct reference in the manuscript.

11. For representative immunostaining images, the panels should indicate synaptic markers (for example Fig. 5).

Answer: We have now indicated in the figure panels the specific pre- and post-synaptic markers used for the array tomography staining.

January 30, 2019

RE: Life Science Alliance Manuscript #LSA-2019-00325-T

Dr. Eloise Hudry
INSERM U986 Université Paris-Sud
Neurology
114 16th street
Orsay 92265
France

Dear Dr. Hudry,

Thank you for transferring your manuscript entitled "OPPOSING ROLES OF APOLIPOPROTEIN E IN AGING AND NEURODEGENERATION" to Life Science Alliance. Your manuscript was previously reviewed at another journal, and the editors transferred those reports to us with your permission. You furthermore provided a point-by-point response to all concerns raised and adapted your manuscript accordingly.

I appreciate your findings and the introduced changes and would thus be happy to publish your paper in Life Science Alliance pending final revisions necessary to meet our formatting guidelines:

- please add a callout to Fig 5A and Fig 5C in the manuscript text (only B and D currently called out)
- please note that we only have 'supplementary figures' at LSA, please modify your EV figures accordingly
- please add the supplementary figure legends to the main manuscript text file and upload the S figures as individual files
- please fill in all mandatory fields in our submission system and sign the electronic license to publish form
- please link your profile in our submission system to your ORCID iD, you should have received an email with instructions on how to do this.

A. FINAL FILES:

-- High-resolution figure, supplementary figure and video files uploaded as individual files: See our

detailed guidelines for preparing your production-ready images, <http://life-science-alliance.org/authorguide>

B. MANUSCRIPT ORGANIZATION AND FORMATTING:

Full guidelines are available on our Instructions for Authors page, <http://life-science-alliance.org/authorguide>

Sincerely,

Andrea Leibfried, PhD
Executive Editor
Life Science Alliance
Meyershofstr. 1
69117 Heidelberg, Germany
t +49 6221 8891 502
e a.leibfried@life-science-alliance.org

February 6, 2019

RE: Life Science Alliance Manuscript #LSA-2019-00325-TR

Dr. Eloise Hudry
Massachusetts General Hospital - Harvard
Neurology
114 16th street
Charlestown, MA 02129

Dear Dr. Hudry,

Thank you for submitting your Research Article entitled "OPPOSING ROLES OF APOLIPOPROTEIN E IN AGING AND NEURODEGENERATION". It is a pleasure to let you know that your manuscript is now accepted for publication in Life Science Alliance. Congratulations on this interesting work.

DISTRIBUTION OF MATERIALS:

Again, congratulations on a very nice paper. I hope you found the review process to be constructive and are pleased with how the manuscript was handled editorially. We look forward to future exciting submissions from your lab.

Sincerely,
